# Effectiveness of LRB in Curved Bridge Isolation: A Numerical Study

**Praveen Kumar Gupta [1], Goutam Ghosh [2], Virendra Kumar [3], Prabhu Paramasivam [4] and Seshathiri Dhanasekaran [5,*]**

1   Department of Civil Engineering, GLA University, Mathura 281406, India
2   Department of Civil Engineering, Motilal Nehru National Institute of Technology Allahabad, Prayagraj 211004, India
3   Department of Mechanical Engineering, Harcourt Butler Technical University, Kanpur 208002, India
4   Department of Mechanical Engineering, College of Engineering and Technology, Mettu University, Mettu 318, Ethiopia
5   Department of Computer Science, UiT The Arctic University of Norway, 9037 Tromso, Norway
*   Correspondence: seshathiri.dhanasekaran@uit.no

**Abstract:** Lead Rubber Bearings (LRBs) represent one of the most widely employed devices for the seismic protection of structures. However, the effectiveness of the same in the case of curved bridges has not been judged well because of the complexity involved in curved bridges, especially in controlling torsional moments. This study investigates the performance of an LRB-isolated horizontally curved continuous bridge under various seismic loadings. The effectiveness of LRBs on the bridge response control was determined by considering various aspects, such as the changes in ground motion characteristics, multidirectional effects, the degree of seismic motion, and the variation of incident angles. Three recorded ground motions were considered in this study, representing historical earthquakes with near-field, far-field, and forward directivity effects. The effectiveness of the bi-directional behavior considering the interaction effect of the bearing and pier was also studied. The finite element method was adopted. A sensitivity study of the bridge response related to the bearing design parameters was carried out for the considered ground motions. The importance of non-linearity and critical design parameters of LRBs were assessed. It was found that LRBs resulted in a significant increase in deck displacement for Turkey ground motion, which might be due to the forward directivity effect. The bi-directional effect is crucial for the curved bridge as it enhances the displacement significantly compared to uni-directional motion.

**Keywords:** curved bridge; lead rubber bearing; bi-directional effect

## 1. Introduction

An LRB is an isolation system that is composed of multiple layers of rubber and steel plates with a central lead core. It is more effective for structural seismic isolation, as reported by Robinson and Tucker [1]. Robinson [2] studied the ability of the lead core to provide hysteresis energy dissipation which can protect the structure from the frequency of earthquake damage. Later, LRBs (Figure 1) were widely used for the isolation of buildings (Robinson and Tucker; Charleson et al.) [3] and bridges (Built; Robinson and Tucker) [4,5]. Kelly [6] reported that lead plug bearings significantly increased damping compared to laminated rubber bearings. However, after the lead plug is inserted, part of the self-centering performance of the laminated rubber bearing is lost. Hirasawa et al. [7] pointed out that the hysteresis loop of the lead rubber bearing is stable under repeated loading. Ghobarah and Ali [8] considered LRBs for seismic isolation of a continuous bridge. The ductility of the bridge pier is compared with the energy dissipation capacity of the LRB. Turkington et al. proposed a design method for LRBs for bridges [9–11]. The main advantage of this method is that it is based on the elastic response spectrum commonly used in the design. Skinner et al. [12]

showed that LRBs have almost no strain rate dependence in a wide frequency range, including typical seismic frequencies. In addition, these have stable behavior under repeated loads, independent of temperature.

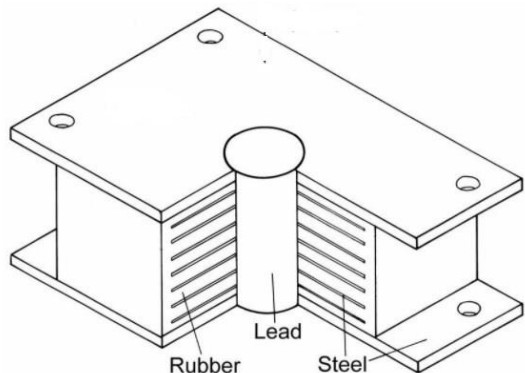

**Figure 1.** Lead Rubber Bearing [1].

Benzoni et al. [13] experimentally studied the effects of axial load and strain rate on the performance of full-size lead-core elastic bearings used in bridges. A numerical model was developed to check bearing response with changes in key performance variables. Choun et al. [14] investigated the effect of changes in the mechanical properties of the LRB on the response of the seismic isolation system. Changes in the mechanical properties of the vibration isolator significantly affected the shear strain of the vibration isolator and the acceleration response of the superstructure. Kumar et. al. [15] concluded with the potential application and future challenges with regards to filler and their polymer composites. Hameed et al. [16] studied the effects of LRBs and ground motion characteristics on the response of isolated bridges. The critical parameters were the ground motion characteristics. The ratio of peak ground acceleration to peak ground speed (PGA/PGV) was represented as the damage index.

In elastomer-based isolation systems, LRBs are critical elements. In this type of bearing, the rubber provides lateral flexibility to prolong the structure's life, whereas the lead core releases energy during the cyclic movement of an earthquake [17]. Wang and Liu [18] established the calculation formula and an appropriate range of design parameters systematically. This comprises the diameter and height of the lead, the effective design deflection, the equivalent linear model, the shear strain of the LRB, the initial stiffness, and the secondary stiffness. Yang and Zhang [19] determined that the axial and shear coupling response of the LRB plays an important role in the safety of an insulated base building. A simple amplification factor of 2.5 was proposed to increase the axial capacity of the LRB when the deformation by shear reached the maximum total displacement. Such a simple amplification factor can result in a low probability of LRB failure during an earthquake. Gang and Guanya [20] concluded that the deterioration of laminated elastomeric bearings and shear keys might play a significant role in evaluating the performance of bridge constructions. Compared to the combined LRB and elastomeric bearing system, the overall design displacement was reduced for all LRBs, as mentioned by Kim and Lee [21]. Chen and Li [22,23] discovered that the LRBs managed the relative displacement between the beams and the bearings, hence reducing the probability of disassembly; however, they cannot suppress the seismic demands of the column. In contrast, when vibrating foundations were used, the dynamic performance of the columns was greatly improved, and it is anticipated that the columns will remain elastic through high seismic excitations. The LRBs could only minimize the seismic inertial force of the superstructure. In contrast, the oscillating base could also isolate the seismic inertial force created by the column, which is crucial for tall column bridges.

Hamaguchi and Kikuchi [24] noted that the torsion of LRBs during biaxial motion does not significantly affect the limit states.

Curved highway bridges are common at urban crossroads and areas with limited space. Due to their irregular geometry, such bridges are often implicated in various vibrational modes, making them more susceptible to vertical components of ground motion. The damage sustained by the Baihua Bridge during the 2008 Wenchuan earthquake in China is a prime example of such bridge damage. This bridge deck comprised many curved (β = 1130 and R = 66 m) and straight parts. The curved segments failed due to the substantial in-plane motion of the deck, whereas the straight segments remained undamaged despite a 60 cm lateral displacement at some of their lateral sliding bearings (Liu and Wang, [25]). Some evidence shows that due to the insufficient seat length, the seismic pounding between the deck segments at in-span joints resulted unseating of the curved segments from their supports. (Kawashima et al. [26]). Ju [27] determined that derailment of trains running on bridges with LRBs and the base shear of bridges with LRBs was significantly lower than those without LRBs. The numerical results of Kabir et al. [28] revealed that long-duration ground vibration dominated both component failure probability and bridge system failure probability which are more than near-field and far-field ground motions.

In the present study, the seismic performance of a continuous curved bridge isolated with LRBs under different seismic loading conditions is investigated. Three different ground motions, viz. near-field (with and without forward directivity) and far-field, have been considered. The effect of the incidence angle of the ground motions has also been investigated.

## 2. Modeling

A horizontally curved reinforced concrete bridge (Figure 2) (with a radius of curvature of 315 m) has been considered for the study (Li Yu et al.) [29]. The bridge consists of 7 spans that comprise 5 intermediate sections of 25 m in length and a final section of 20 m. The total length of the bridge is 165 m. The superstructure consists of a single chamber box girder of a constant section with an area of 3.099 m$^2$ and a longitudinal bending moment of inertia of 0.599 m$^4$. M40-grade concrete has been considered. All the piers are made of M30 grade concrete, with a height of 11 m and a circular cross-sectional area of 1.767 m$^2$. Spiral reinforcement (20 Φ) is used as stirrups with a pitch of 10 cm, and longitudinal reinforcement bars of 15 nos. 28 Φ are used together with a cover layer thickness of 5 cm. The bridge was modeled using the finite element program in SAP2000 [30] software. The bridge superstructure and piers were modeled using beam elements, and masses were concentrated at discrete points.

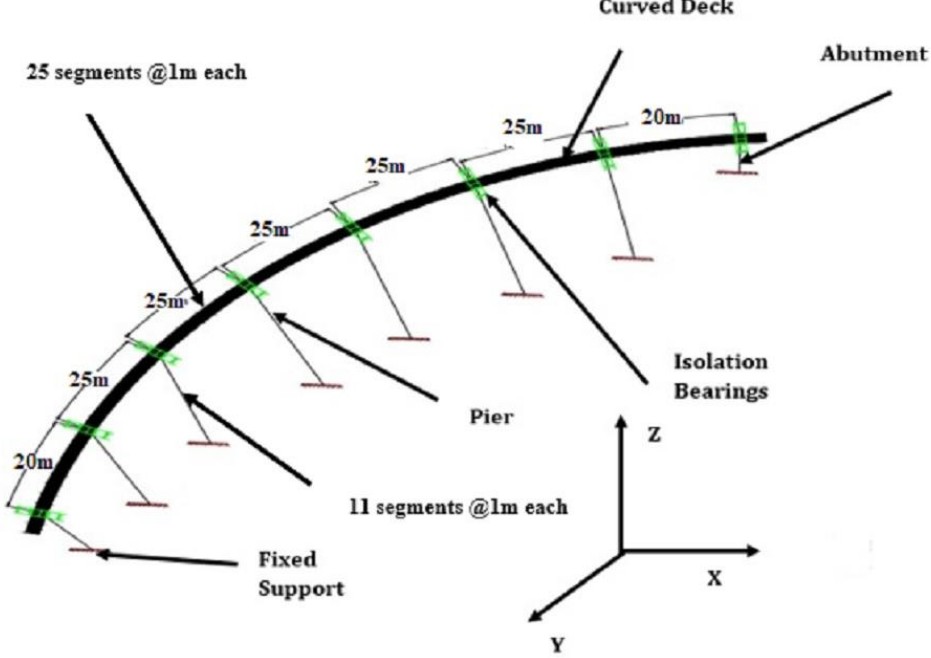

**Figure 2.** Curved bridge considered for the study.

The piers are considered to be fixed as they are considered to be resting on the rock. The abutment is assumed to be rigid. LRBs have been modeled as link elements (Figure 3). The force-deformation behavior of LRB is shown in Figure 4a,b. The bridge is supported by roller bearings at the abutment and fixed to the bottom of the pier. For each case, two LRBs are considered at each abutment location, and four LRBs are considered at each pier location. In the model, the global longitudinal (X) axis is the straight line connecting two abutments, and the global transverse (Y) axis is the perpendicular horizontal direction with respect to the X axis. The vertical axis is the global Z direction.

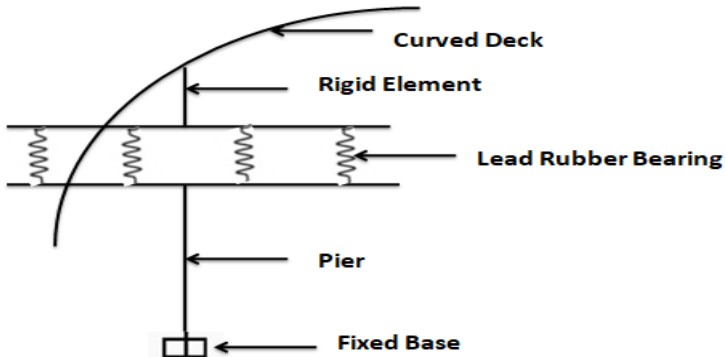

**Figure 3.** Modeling of isolation bearings, pier, and bridge deck junction.

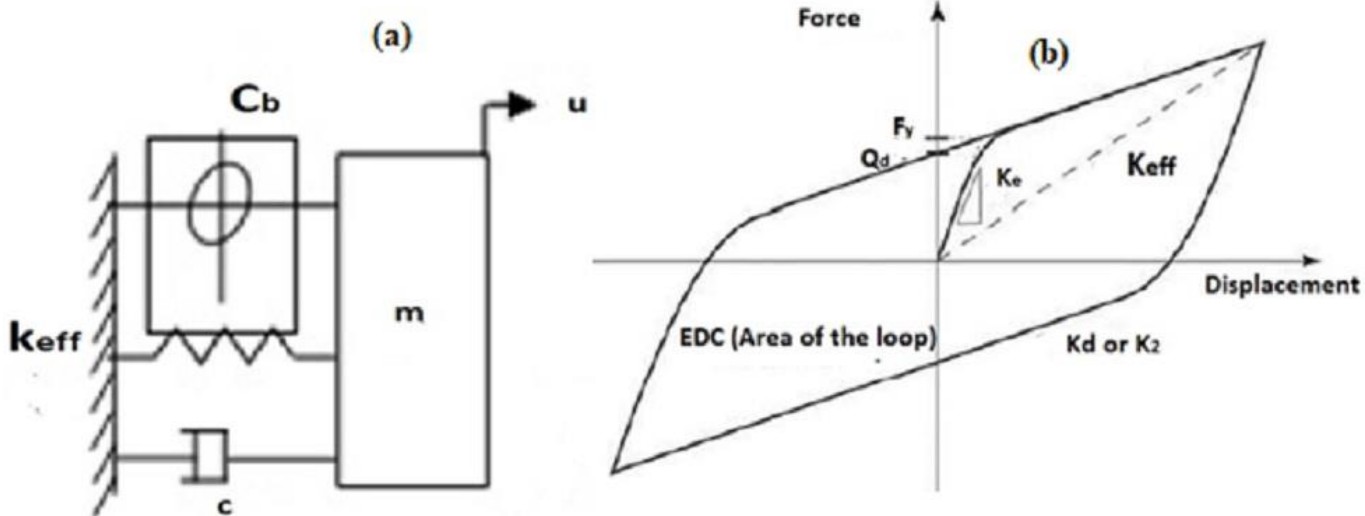

**Figure 4.** Lead rubber bearing (**a**) analytical model, (**b**) force-deformation behavior.

For gravity loads, static analysis is typically adopted. Non-Linear Time History Analysis (NLTHA) was used to analyze the seismic loads of bridge structures with different types of bearings. Since the bridge is curved, the resultant of the responses in the X and Y directions were considered. The lumped plasticity model was considered in the study. In the finite element model, the mass is concentrated at discrete points between segments (elements) automatically generated by SAP2000 [30]. The SAP2000 program automatically considers mass distribution. The program automatically calculates the translational mass of all elements in the global direction of the bridge and assigns them as the concentrated mass at each node according to the length of the tributaries. In order to approximate the distribution of mass with concentrated mass, it is necessary to define a sufficient number of nodes and segments (units). According to ATC 32 [31], it is recommended that each superstructure span has at least 4 segments and piers have at least 3 segments. In this study, 11 types of elements have been considered for piers. The superstructure (deck) spans have been modeled by 20 elements for a 20-m span and 25 elements for a 25-m span. Table 1 mentions

the details of the elements considered in each part of the bridge. Non-linearity in the pier was considered by plastic hinges, and the hinge properties were considered using a concentrated plastic hinge model according to FEMA 356 [32].

**Table 1.** Details of the number of elements considered in bridge modeling.

| Location | Span/Height (m) | Nos. of Segments (Elements) Considered | Length of Each Element (m) |
|---|---|---|---|
| Superstructure | 20 | 20 | 1 |
| | 25 | 25 | 1 |
| Pier | 11 | 11 | 1 |

The Bouc–Wen model [33] was used in this study and is shown in Figure 4. The model is presented by three parameters: (a) Post Yield Stiffness ratio $k_2$ or $k_d$; (b) Characteristic strength $Q_d$ inter concept of zero displacements; (c) Yield strength $f_{py}$. The Characteristic Strength $Q_d$ of LRB is provided by the yield strength of lead in shear $f_{py}$ and cross-sectional area of the lead plug $A_p$, as given by the following equation.

$$Q_d = f_{py} \times A_p \tag{1}$$

For specified design displacement, $D$, Post-yield stiffness, $k_2$, the effective stiffness, $K_{eff}$ of LRB is given by

$$K_{eff} = k_2 + \frac{Q_d}{D} \tag{2}$$

The effective damping $\beta_{eff}$ of Bilinear system for the yield displacement, $D_y$, is given by

$$\beta_{eff} = \frac{4(D - D_y)Q_d}{2\pi K_{eff}D^2} \tag{3}$$

The coupled force-deformation relationship is given by

$$F_1 = R_1 K_{e1} u_1 + (1 - R_1)F_{y1}z_1 \tag{4}$$

$$F_2 = R_2 K_{e2} u_2 + (1 - R_2)F_{y2}z_2 \tag{5}$$

where, $K_{e1}$ and $K_{e2}$ are the elastic stiffnesses in two orthogonal horizontal directions, $F_{y1}$ and $F_{y2}$ are the yield forces and $u_1$ and $u_2$ are displacements in the two directions, $R_1$ and $R_2$ are the ratios of post-yield stiffnesses to elastic stiffnesses and $Z_1$ and $Z_2$ are internal hysteretic variables which have a range of $\pm 1$, with the yield surface as represented by $\sqrt{Z_1^2 + Z_2^2} = 1$.

As per the Park et al. model [34,35], the hysteretic dimensionless quantity $Z$ is governed by the following Equation (6).

$$ZU^y = \left\{ A - |Z|^\eta \left( \gamma Sign\left( \dot{U}Z \right) + \beta \right) \right\} \dot{U} \tag{6}$$

where $\gamma$, $\beta$, $\eta$, and $A$ are dimensionless quantities that control the shape of the hysteresis loop, $\dot{U}$ stands for the velocity. In the model, $Z_x$ and $Z_y$ are the hysteretic dimensionless quantities, which are governed by the following coupled differential equations that account for the interaction between the $X$ and $Y$ directions. The paper represents $Z_x$ and $Z_y$ by $Z_1$ and $Z_2$, respectively.

$$\begin{Bmatrix} Z_x U_x^y \\ Z_y U_y^y \end{Bmatrix} = AI \begin{Bmatrix} \dot{U}_x \\ \dot{U}_y \end{Bmatrix} - \begin{pmatrix} Z_x^2 \left( \gamma Sign\left( \dot{U}_x Z_x \right) + \beta \right) & Z_x Z_y \left( \gamma Sign\left( \dot{U}_y Z_y \right) + \beta \right) \\ Z_x Z_y \left( \gamma Sign\left( \dot{U}_x Z_x \right) + \beta \right) & Z_y^2 \left( \gamma Sign\left( \dot{U}_y Z_y \right) + \beta \right) \end{pmatrix} \begin{pmatrix} \dot{U}_x \\ \dot{U}_y \end{pmatrix} \tag{7}$$

$U_x^y$ and $U_y^y$ are the yield displacements in $X$ and $Y$ direction, respectively.

$\dot{U}_x$ and $\dot{U}_y$ are the velocity components in *X* and *Y* directions, respectively.

Additional models, typically adopted for LRBs, are the ones described in [36], obtained on the basis of the uniaxial model proposed in [37], and a more recent one [38] obtained by extending a previous efficient model [39,40].

## 3. Parametric Study

The design of LRBs is based on different standards provided in various specifications (AASHTO 1999, FEMA 356, IRC 2015) [41,42] and literature (Roeder and Stanton, Priestley et al., Dolce et al.) [43–45]. The design of LRB can be affected by the isolated time period, damping ratio, and type of seismic loading. In turn, these depend on the size and design of the LRB. The sensitivity of the bridge response has been studied for 1.5 to 3 s time periods and 0.10 to 0.25 damping ratios, according to the guidelines available in the literature (Ersoy et al., Huanding et al., FEMA-356) [46,47].

Three recorded ground motions from the PEER Strong Earthquake Database (PEER 2014) [48] have been considered without scaling viz. (1) Imperial Valley (1940), with perpendicular components (PGA 0.313 g and PGA 0.215 g), (2) Kobe (1995), with two earthquake components (PGA 0.83 g and PGA 0.63 g), and (3) Turkey (1992), as indicated in Table 2 and Figure 5. Figure 6 shows the response spectrum curve considered in the study. The conditions close to far-field, near-field, and forward directivity effects, respectively, were represented by the selected ground motions.

**Table 2.** Earthquake-recorded motions considered.

| Record | Event | Magnitude | Station | Orientation | PGA (g) | Distance to Fault (km) |
|--------|-------|-----------|---------|-------------|---------|------------------------|
| 1 | Imperial Valley (1940) | 7.0 | 117 El Centro Array #9 | IMP VALL/ I-ELC180 | 0.31 | 8.3 |
| 2 | Imperial Valley (1940) | 7.0 | 117 El Centro Array #9 | IMP VALL/ I-ELC270 | 0.21 | 8.3 |
| 3 | Kobe (1995) | 6.9 | KJMA | RSN1106_ KOBE_KJM000 | 0.83 | 0.96 |
| 4 | Kobe (1995) | 6.9 | KJMA | RSN1106_ KOBE_KJM090 | 0.63 | 0.96 |
| 5 | Turkey (1992) | 6.9 | Erzincan | Erzincan, EW | 0.50 | 4.3 |
| 6 | Turkey (1992) | 6.9 | Erzincan | Erzincan, NS | 0.39 | 4.3 |

The time history analysis was performed using the numerical integration technique Hilber-Hughes-Taylor (HHT), where the value of the parameter $\alpha$ is specified as $-0.33$. The sectional behavior of different elements has been controlled by considering the non-linear force and deformation behavior. In this study, the seismic isolation bearing was first designed for the selected earthquake, and then the isolated bridge performance was determined. Appendix A (Tables A1–A3) shows the different seismic loads (uni-directional and bi-directional) considered for ground motions corresponding to the Imperial Valley, Kobe, and Turkey earthquakes.

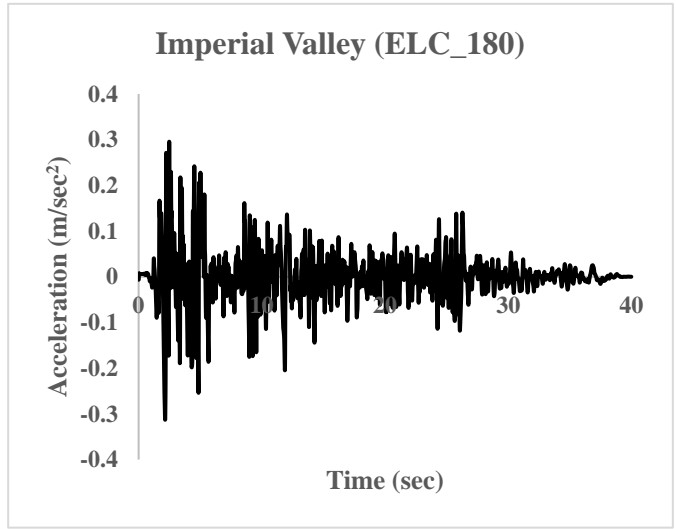
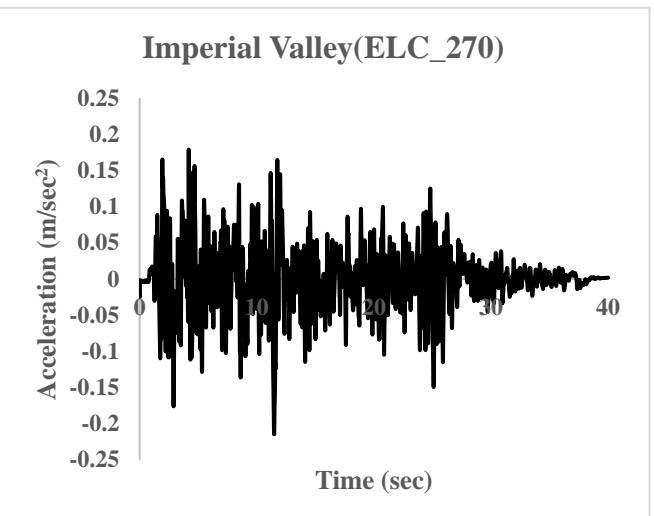
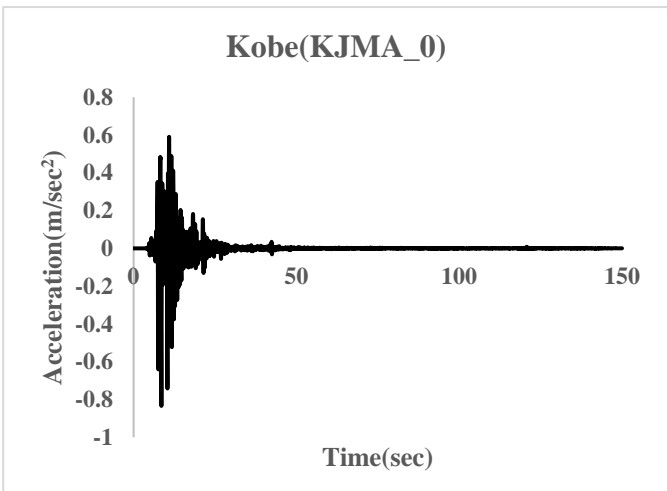
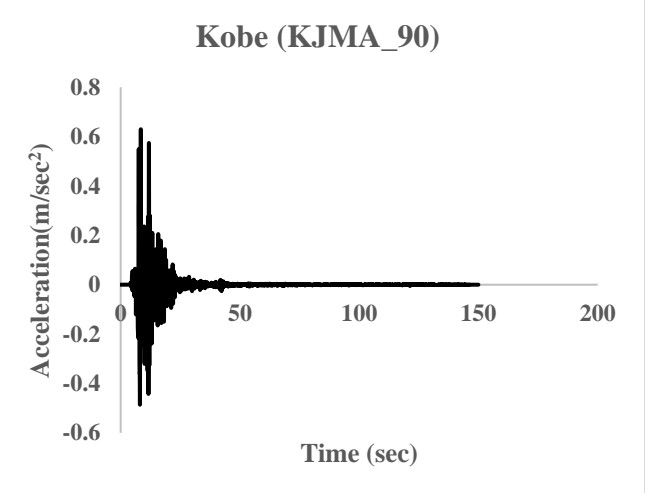
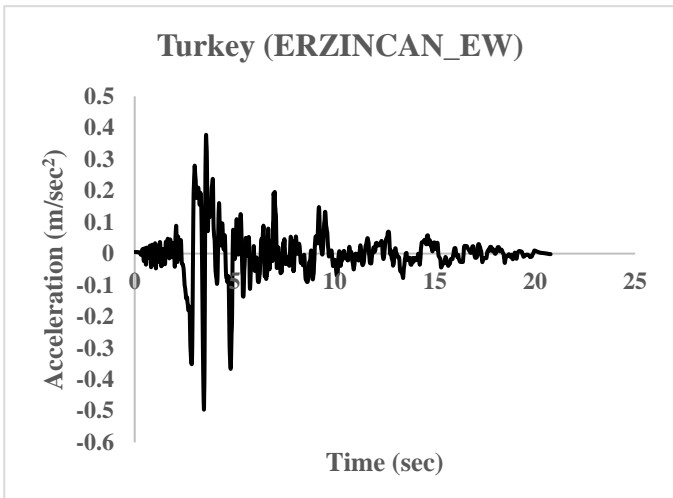
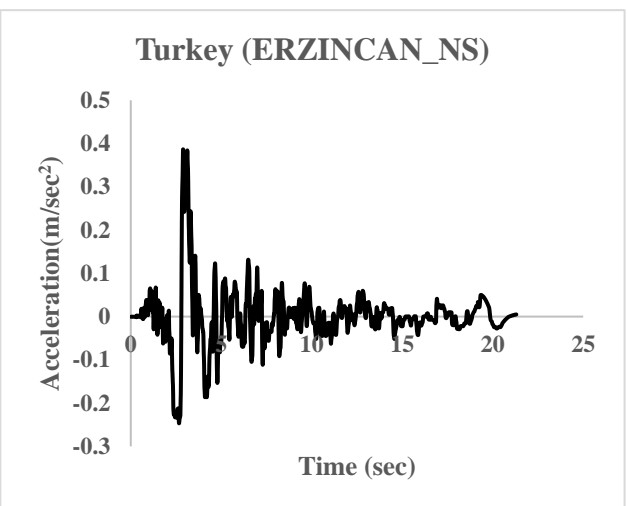

**Figure 5.** Selected recorded earthquake motion time histories.

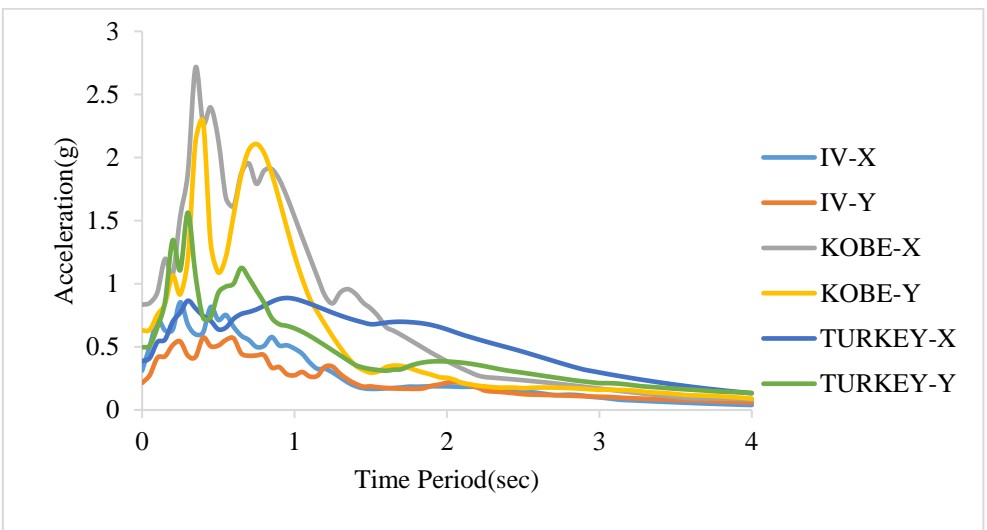

**Figure 6.** Response spectrum of Imperial Valley (IV), Kobe, and Turkey ground motions.

## 4. Effect of Incidence Angle

The earthquake's incident angle ($\alpha$) is an important consideration because the bridge is of a curved configuration and the contribution of responses from both horizontal directions is vital for the bearing design. This study considered the key response parameters, such as deck acceleration, bearing displacement, force transmitted to the pier, and pier torsional moment.

Figure 7a–l depict the bridge response to the uni-directional ground motion for the Imperial Valley, Kobe, and Turkey earthquakes, with incident angles ranging from 0° to 90°. It was observed that the variations of response parameters with respect to the earthquake incidence angle were non-linear. For all the considered ground motions, the change in bearing displacement was small (up to 5%). The maximum deck acceleration is decreased for all the motions, especially between 45° and 67.5°. The maximum reduction of deck acceleration was observed for Kobe ground motion (about 13.16%). The variation of force transmitted to the pier was small (0–4%) for the Imperial Valley and Kobe earthquakes. In the case of the Turkey earthquake, however, the force transmitted to the pier was reduced by about 8%. The variation of the pier torsional moment was almost negligible in the Imperial Valley earthquake, whereas the variations were different for other ground motions. In the case of the Kobe earthquake, the maximum reduction of pier torsional moment (about 33%) was observed for 45° incidence angle, while for 90° incidence angle, the variations were almost insignificant. In the case of Turkey's earthquake, the maximum reduction of pier torsional moment (about 14%) was observed for higher incidence angles of 67.5°–90°.

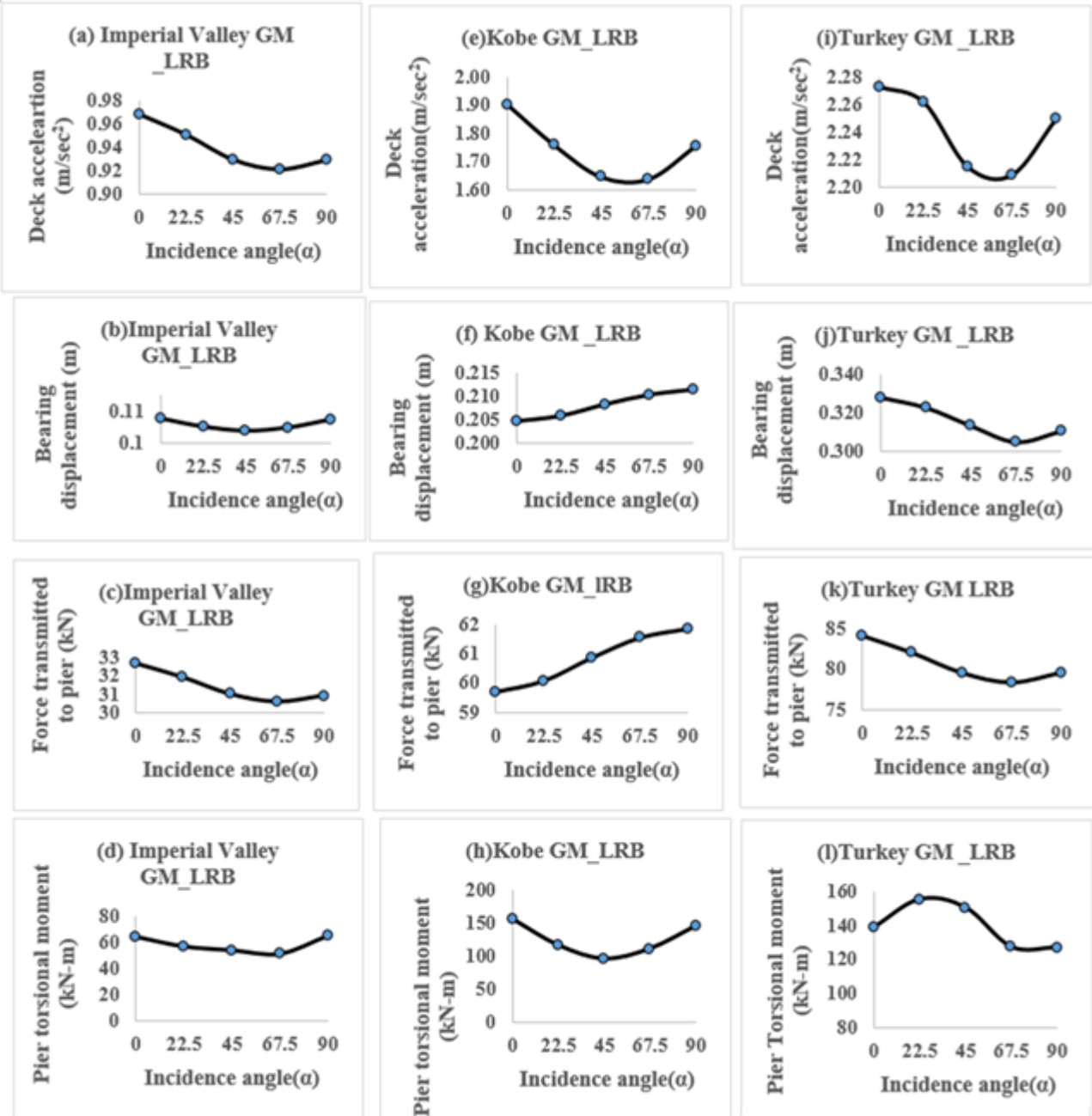

**Figure 7.** The bridge responses for uni-directional Imperial Valley (**a–d**), Kobe (**e–h**), and Turkey (**i–l**) Ground Motions with the variation of incidence angle.

Figure 8a–l depict the bridge response for bi-directional Imperial Valley, Kobe, and Turkey earthquakes at various incident angles ranging from 0° to 90°.

In the case of bi-directional motion, the variation of deck acceleration was almost insensitive (0–3%) for the Imperial Valley and Turkey earthquakes. In contrast, for the Kobe earthquake, the same was increased by about 11% at an incidence angle of 90°. The variation of bearing displacement and force transmitted to the pier was minimal (0–3%) in all the earthquake motions.

Pier torsional moment variation increases with the increase in incidence angle for all the considered ground motions, with the maximum value occurring at an incidence angle of 90°. However, in all cases, the pier torsional moment values are insignificant.

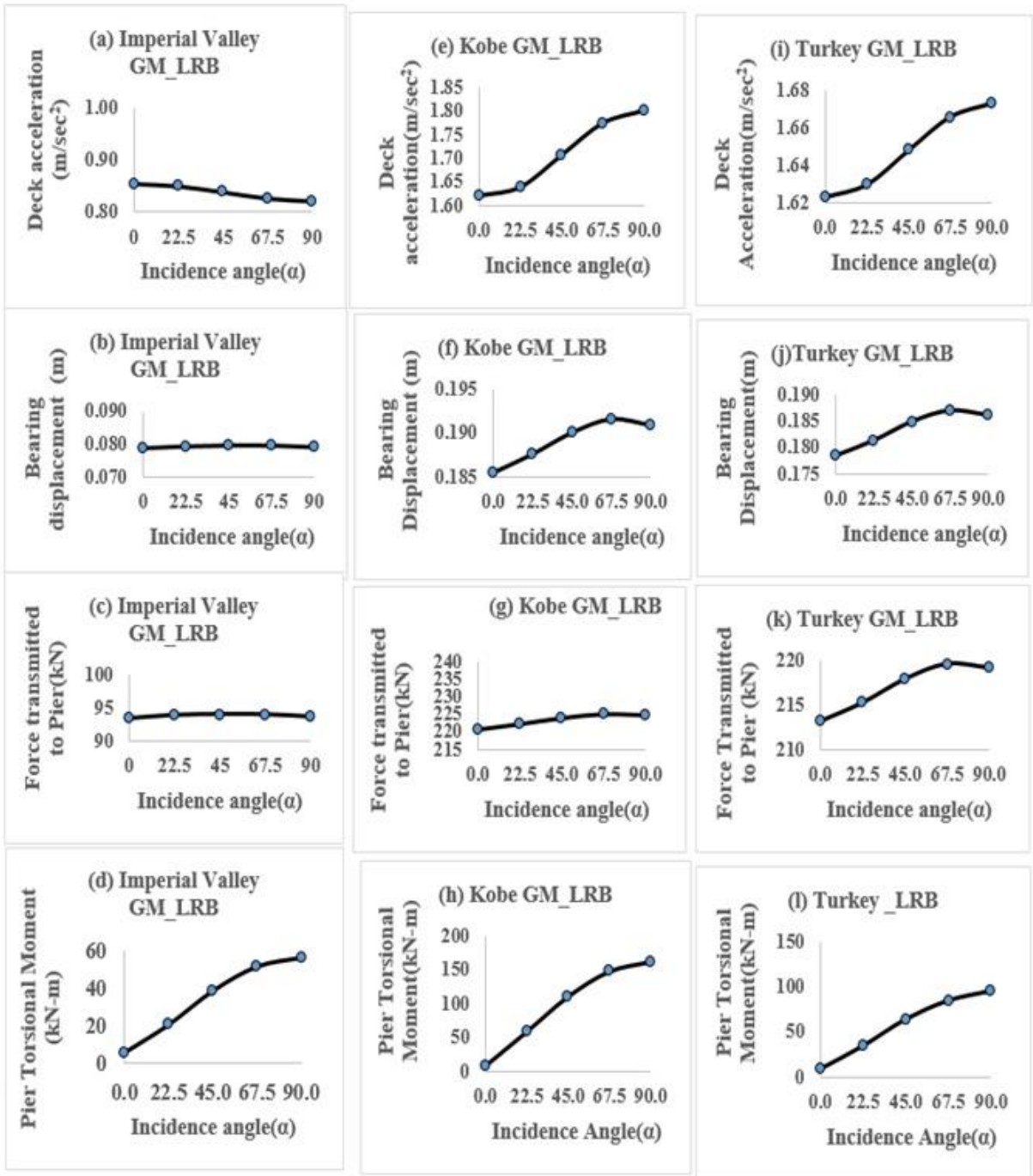

**Figure 8.** Effect of incidence angle (α) on response parameter of the isolated bridge under bi-directional Imperial Valley (**a–d**), Kobe (**e–h**), and Turkey (**i–l**) Ground Motions.

## 5. Sensitivity of Response Parameters

In the sensitivity analysis, the changes in the peak response of the bridge within the selected range of the bearing design parameters were determined.

### 5.1. Uni-Directional Loading

Figure 9a–i show the bridge response variation with the variation of damping ratio for the considered time period ranges for uni-directional Imperial Valley, Kobe, and Turkey earthquakes, respectively. It can be seen from the figures that most of the response parameters are non-linear with respect to changes in damping ratios. A higher damping ratio reduces the bearing displacement for all the considered earthquakes. The yield force

of the seismic isolation bearing increases as the damping ratio increases. The reduction in bearing displacement was observed in Imperial Valley (15–20%) and Turkey (8–19%) ground motions, especially in higher time periods exceeding 1.5 s. For Kobe ground motion, it was found that the bearing displacement changed very little relative to the damping ratio (3–5%). It was also found that in the case of Turkey ground motion, the variation of bearing displacement was linear and steeper for the higher time periods ranging from 2.5 to 3.0 s which might be due to the forward directivity effect that causes a sharp decrease in the bearing displacement for higher time periods.

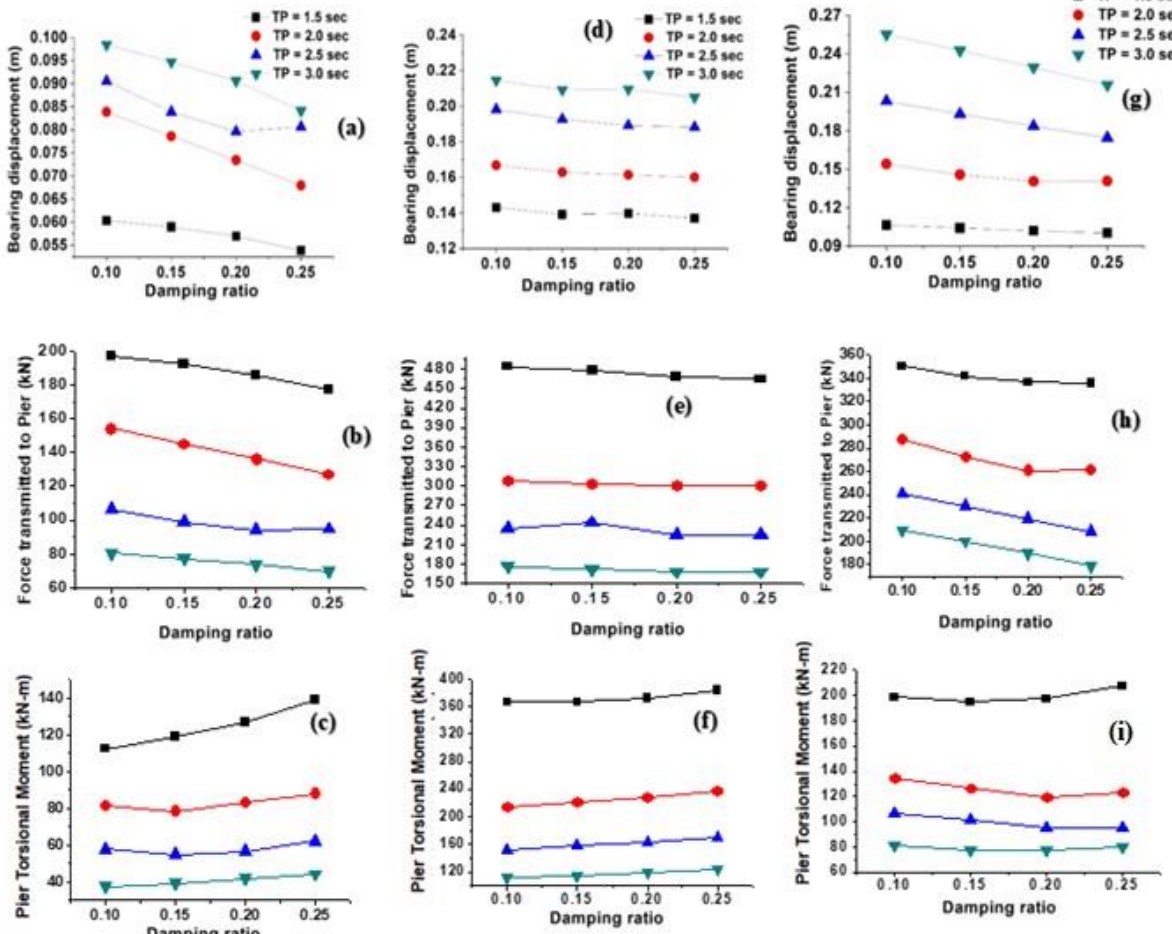

**Figure 9.** The Bridge Response sensitivity with respect to the damping ratio of LRBs under uni-directional (i) Imperial Valley (**a–c**), (ii) Kobe (**d–f**), and (iii) Turkey (**g–i**) ground motions.

It was observed that the force transmitted to the pier was not sensitive (0–4%) to the damping ratio for Kobe ground motion. In the case of Imperial Valley, the force transmitted to the pier was sensitive to the damping ratio (10–19% reduction) in the lower time period ranges of 1.5 to 2.0 s. However, in the case of Turkey ground motions, the force transmitted to the pier was sensitive (9–17% decrease) with respect to the damping ratio, especially for higher time periods from 2 to 3 s.

In the case of Imperial Valley ground motion, pier torsional moments were found to be more sensitive with respect to damping ratio for a 1.5 s time period, while for time periods from 2.0 to 3.0 s, the same was found to be almost insensitive and found to be in the range of 5–10%. In the case of Kobe and Turkey ground motions, the pier torsional moment was found to be small (0–8%).

Figure 10a–i show the variation of the bridge response for variation of the time period with all considered damping ratios for uni-directional Imperial Valley, Kobe, and Turkey

earthquakes, respectively. It was found that with respect to the time period, the response parameters are more sensitive for all the considered damping ratios.

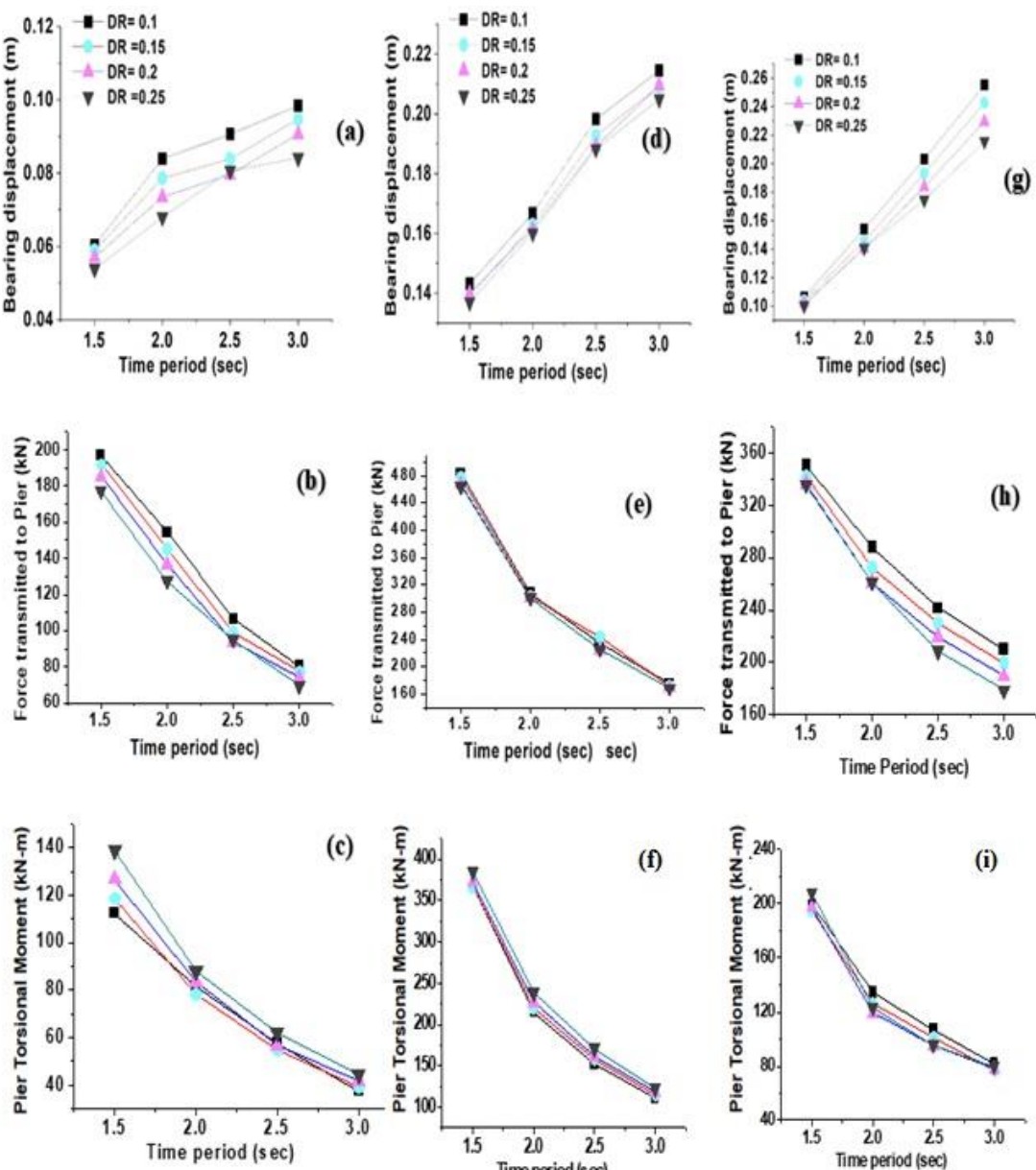

**Figure 10.** Response sensitivity of the isolated bridge with respect to the time period of LRB under uni-directional (i) Imperial Valley (**a**–**c**), (ii) Kobe (**d**–**f**), and (iii) Turkey (**g**–**i**) ground motions.

The bearing displacements increased with respect to the time period for all the considered ground motions. The increased time period imparted additional flexibility to the structure, which increased the bearing displacement. It was observed that in the case of the Imperial Valley and Kobe earthquakes, the increase in bearing displacement is non-linear, while for Turkey's ground motion, the same is almost linear, which might be because of the forward directivity effect.

The transmitted force to the pier was found to be decreased significantly with respect to the increase in the time period for all the ground motions. The maximum decrease (about 67%) in force transmitted to the pier was observed for Kobe's ground motion. However, for Kobe's ground motion, the variations are almost the same for all the considered damping ratios of the bearings. For the Imperial Valley and Turkey earthquakes, the maximum decrease in force transmitted to the pier is about 60% and 47%, respectively.

Pier torsional moment is sensitive with respect to the increase in the time period. The maximum decrease (about 74%) in pier torsional moment was found for Kobe's ground motion. In the case of Imperial Valley and Turkey motions, the decrease in pier torsional moment is about 68% and 60%, respectively. However, the variation of pier torsional moment is almost insensitive for all the damping ratios, especially for Kobe and Turkey ground motions.

### 5.2. Bi-Directional Loading

Figure 11a–i depict the variation in bridge response for different damping ratios over all time periods for bi-directional Imperial Valley, Kobe, and Turkey ground motions, respectively. Similar to the uni-directional cases, the variation of responses with respect to the damping ratio is non-linear. The bearing displacement was decreased with an increase in the damping ratio. In Kobe and Turkey ground motions, the maximum decrease in bearing displacement was about 16% and 11%, respectively, for higher time periods from 2.5 to 3 s. In the case of Imperial Valley ground motion, the maximum variation (about a 16% decrease) was observed for a period of 2.0 s. For all the considered ground motions, the variation of bearing displacement with respect to the damping ratio was found to be almost the same at time periods of 1.5 s. Additionally, a steep variation of bearing displacement with respect to the damping ratio, as observed for the uni-directional case for Turkey ground motion, was reduced in bi-directional ground motions.

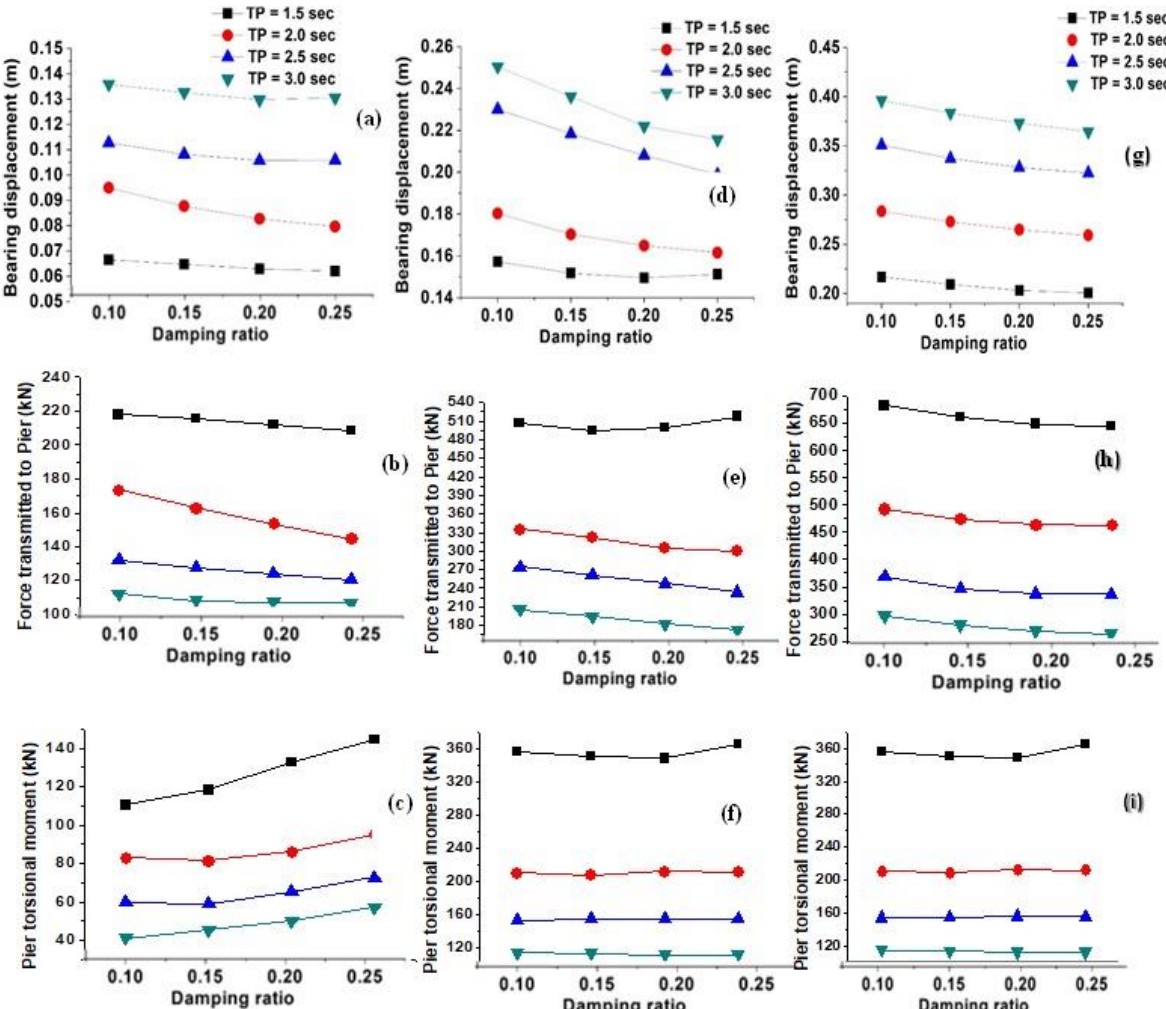

**Figure 11.** Response sensitivity of the isolated bridge with respect to damping ratio of LRB for bi-directional (i) Imperial Valley (**a**–**c**), (ii) Kobe (**d**–**f**), and (iii) Turkey (**g**–**i**) ground motions.

The variation of force transmitted to the pier is small (variations are 6–9%) with respect to the damping ratio for the Turkey ground motion in the time period range from 1.5 to 2.5 s. In the case of the time period of 3.0 s, the variation of force transmitted to the pier is about 13%. In the case of Kobe ground motion, the variation of force transmitted to the pier is found to be insensitive with respect to the damping ratio for the time period of 1.5 s, while for a 2 to 3 s time period, the variation is about 13–19%. In the case of Imperial Valley ground motion, the force transmitted to the pier is found to be sensitive (about 14% variation) with respect to the damping ratio corresponding to the 2 s time period. In contrast, for other time periods, the variation is almost insensitive.

The variation of pier torsional moments is found to be insensitive to the Kobe and Turkey ground motions. The maximum variation (about 21%) of pier torsional moment with respect to damping ratio was observed for Imperial Valley ground motion for a 1.5 s time period.

For bi-directional Imperial Valley, Kobe, and Turkey earthquakes, Figure 12a–i show the variation of the bridge response for varying time periods with all the considered damping ratios. Similar to uni-directional responses, the response parameters are more sensitive to the time period for all the damping ratios.

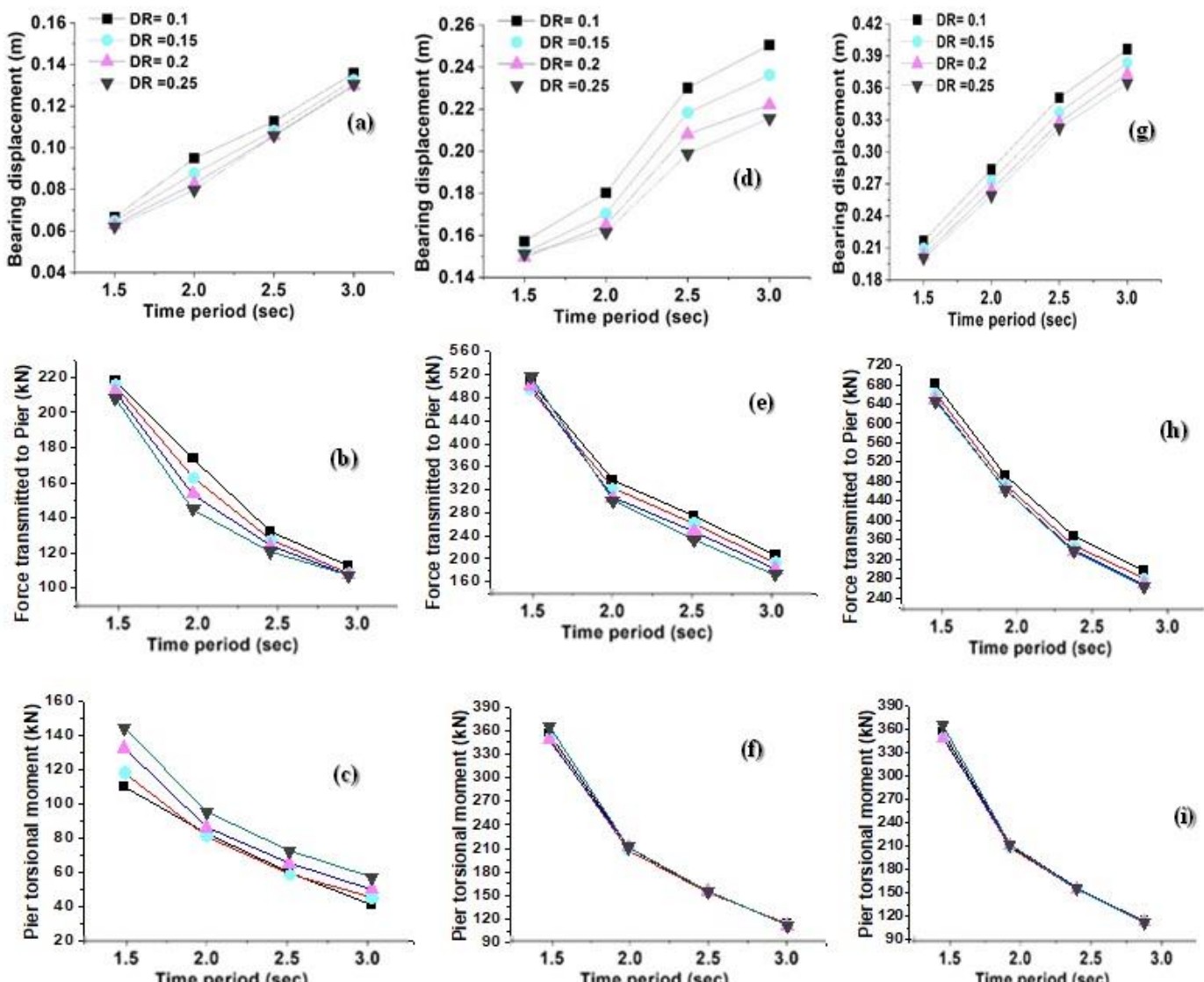

**Figure 12.** Response sensitivity of the isolated bridge with respect to the time period of LRB for bi-directional (i) Imperial Valley (**a–c**), (ii) Kobe (**d–f**), and (iii) Turkey (**g–i**) ground motions.

The bearing displacements are increased with respect to the time period for all the ground motions. The maximum increase in bearing displacement (about 125%) was observed for Imperial Valley ground motion, whereas, for Kobe and Turkey ground motions, the maximum increase was 56% and 85%, respectively. Moreover, the increase in bearing displacement is close to linear and steeper for Turkey's ground motion.

The force transmitted is found to be decreased for every increase in the time period for all the ground motions. The maximum increase in force transmitted (about 69%) has been observed for Kobe ground motion, whereas, for Imperial Valley and Turkey ground motions, the maximum increase is 50% and 62%, respectively. Also, in Kobe and Turkey ground motions, the variations of force transmitted to the pier are found to be almost the same for all the considered damping ratios of the bearings. The variation of the pier torsional moment is non-linear and sensitive to all the ground motions. For Imperial Valley, Kobe, and Turkey ground motions, the pier torsional moment is decreased by about 63%, 72%, and 70%, respectively. However, the variations in pier torsional moment are nearly identical for all damping ratios for Kobe and Turkey ground motions.

## 6. Non-Isolated vs. Isolated Bridge Response

Figure 13 shows the peak response of the non-isolated and isolated bridges for uni-directional and bi-directional Imperial Valley, Kobe, and Turkey ground motions. The various cases are shown in Table 3.

**Table 3.** Various cases considered for comparison of the response of bridge with respect to non-isolated condition.

| Case 1 | **Non-Isolated Bridge** | |
| --- | --- | --- |
| | **Isolated Bridge** | |
| | **Isolation Period (sec)** | **Damping Ratio (%)** |
| Case 2 | 1.5 | 10 |
| Case 3 | 1.5 | 15 |
| Case 4 | 1.5 | 20 |
| Case 5 | 1.5 | 25 |
| Case 6 | 2.0 | 10 |
| Case 7 | 2.0 | 15 |
| Case 8 | 2.0 | 20 |
| Case 9 | 2.0 | 25 |
| Case 10 | 2.5 | 10 |
| Case 11 | 2.5 | 15 |
| Case 12 | 2.5 | 20 |
| Case 13 | 2.5 | 25 |
| Case 14 | 3.0 | 10 |
| Case 15 | 3.0 | 15 |
| Case 16 | 3.0 | 20 |
| Case 17 | 3.0 | 25 |

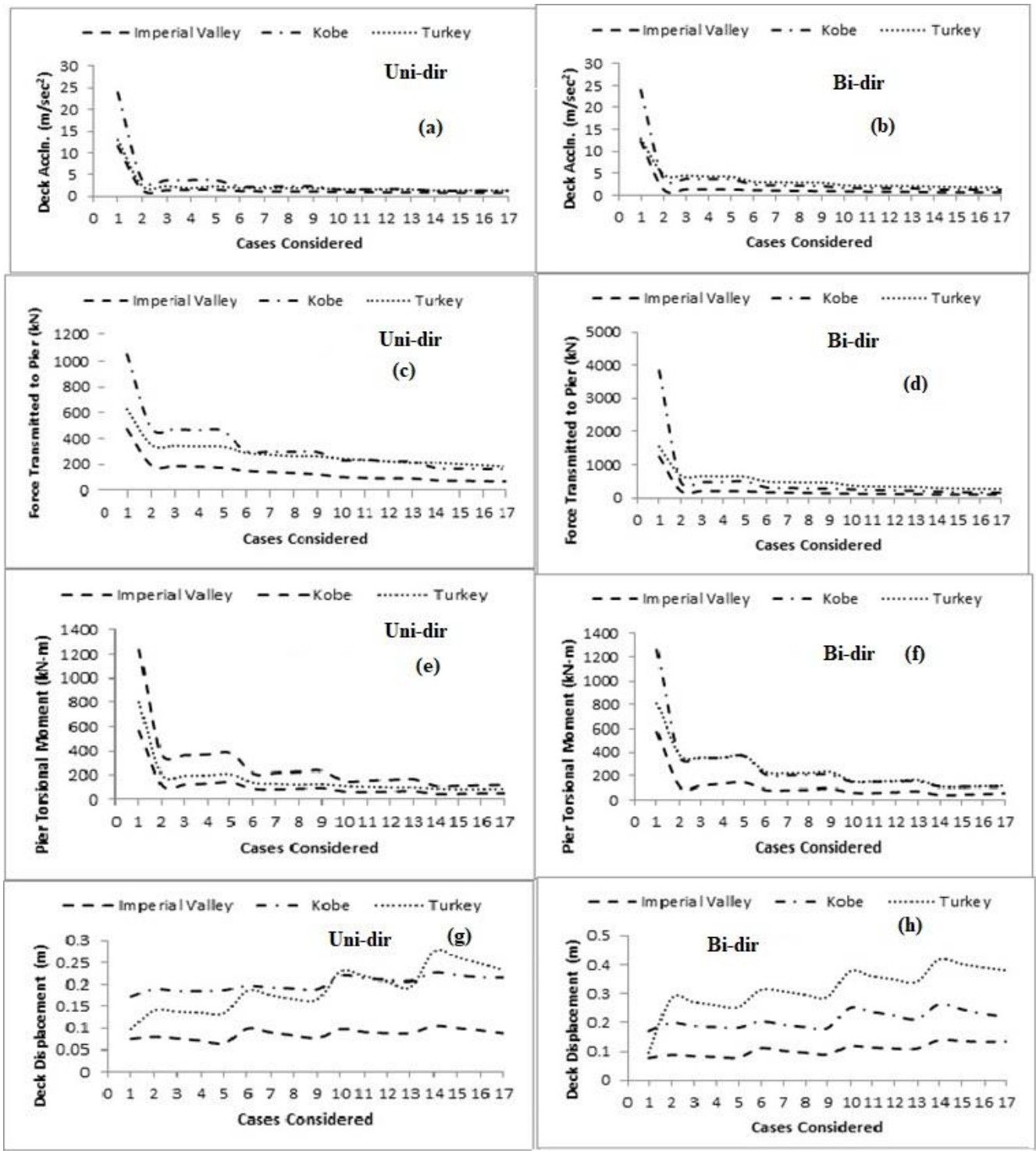

**Figure 13.** Peak response of the bridge (i) Deck Acceleration (**a**,**b**), (ii) Force Transmitted to Pier (**c**,**d**), (iii) Pier Torsional Moment (**e**,**f**) Deck Displacement (**g**,**h**) for Imperial Valley, Kobe, and Turkey earthquakes.

It was found that the LRBs decreased the deck acceleration, the force transmitted to the pier, and the pier torsional moment significantly more than the non-isolated condition for uni-directional and bi-directional cases for all the considered earthquake motions. The LRBs showed their major importance in curved bridges as they significantly reduced the pier torsional moment for three different variations of earthquake motions. This fact

has been noticed in various cases with combinations of time periods and damping ratios of the bearings.

Considering all the cases, for uni-directional Imperial Valley ground motion, it was found that the maximum deck acceleration, the force transmitted to the pier, and pier torsional moment were reduced by about 88–93%, 58–85%, and 80–92%, respectively, than the non-isolated bridge. In the case of uni-directional Kobe ground motions, the maximum deck acceleration and force transmitted to the pier and pier torsional moment were reduced by about 83–94%, 54–84%, and 71–90%, respectively, than the non-isolated bridge.

For bi-directional Imperial Valley ground motion, the maximum deck acceleration force transmitted to the pier and pier torsional moment were reduced by about 87–93%, 82–93%, and 80–92%, respectively, than the non-isolated bridge. While for bi-directional Kobe ground motion, the maximum deck acceleration and force transmitted to the pier and pier torsional moment were reduced by about 83–94%, 86–95%, and 70–90%, respectively, than the non-isolated bridge.

However, in the case of Turkey ground motion, the reduction of the maximum deck acceleration and force transmitted to the pier and pier torsional moment was less, around 80–89%, 44–71%, and 75–90%, respectively, for uni-directional motion and 62–85%, 56–83%, and 53–85%, respectively, for bi-directional motion.

In the case of the isolated bridge, the deck displacement increased more than in non-isolated cases for all the considered earthquakes. However, the maximum increase in deck displacement, for both uni-directional and bi-directional cases, was found for the Turkey earthquake, especially at higher time periods, which is an essential fact from the designer's point of view.

## 7. Bi-Directional Loading Effects

Figure 14 shows the peak response of the isolated bridge for various ground motions. The effect of bi-directional ground motion was found to be significant for the considered earthquake motions. The various cases are shown in Table 4.

**Table 4.** Different cases considered for the Isolated Bridge.

| Cases Considered | Isolated Bridge | |
|:---:|:---:|:---:|
| | Isolation Period (sec) | Damping Ratio (%) |
| Case 1 | 1.5 | 10 |
| Case 2 | 1.5 | 15 |
| Case 3 | 1.5 | 20 |
| Case 4 | 1.5 | 25 |
| Case 5 | 2.0 | 10 |
| Case 6 | 2.0 | 15 |
| Case 7 | 2.0 | 20 |
| Case 8 | 2.0 | 25 |
| Case 9 | 2.5 | 10 |
| Case 10 | 2.5 | 15 |
| Case 11 | 2.5 | 20 |
| Case 12 | 2.5 | 25 |
| Case 13 | 3.0 | 10 |
| Case 14 | 3.0 | 15 |
| Case 15 | 3.0 | 20 |
| Case 16 | 3.0 | 25 |

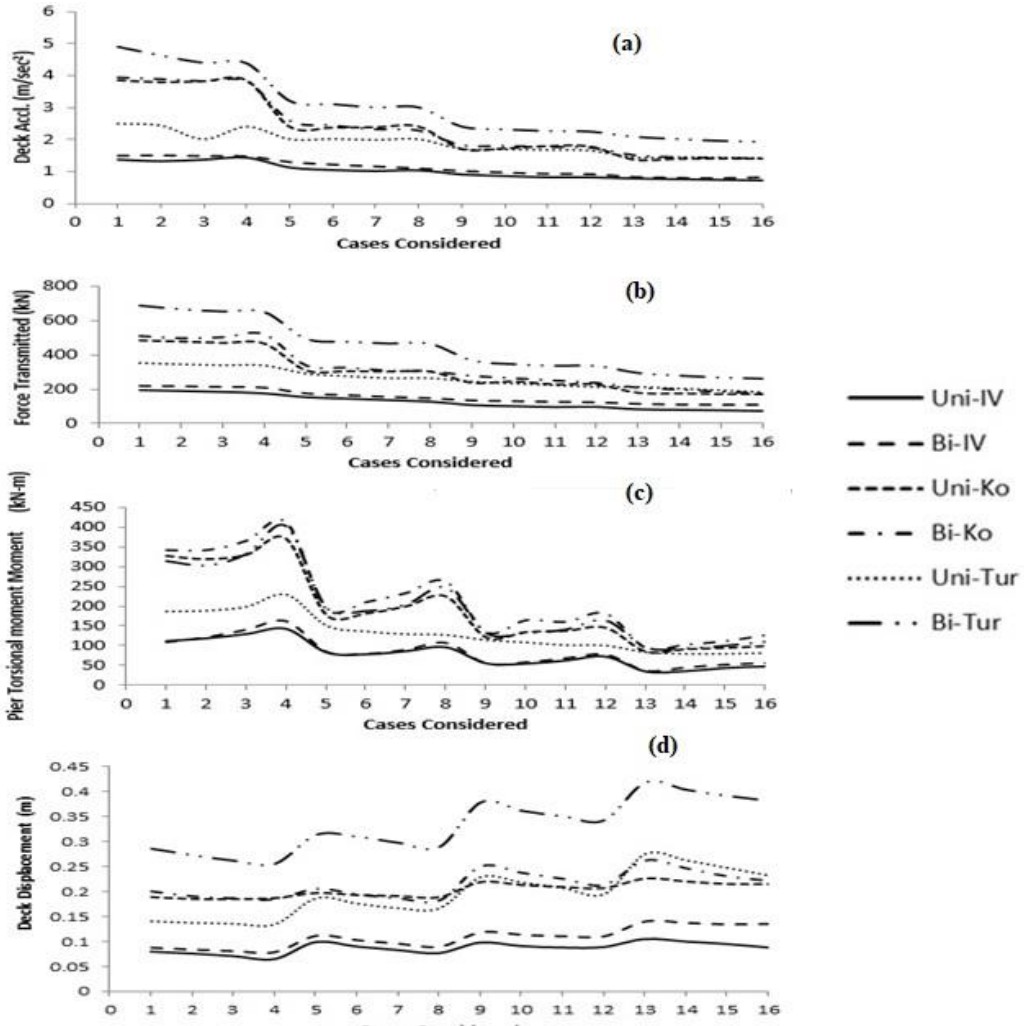

**Figure 14.** The isolated curved bridge response parameters (**a**) Deck Acceleration, (**b**) Force transmitted to Pier, (**c**) Pier Torsional Moment, (**d**) Deck Displacement for Uni-directional (Uni) and Bi-directional (Bi) Imperial Valley (IV), Kobe (Ko) and Turkey (Tur) Ground Motions.

The maximum effect of bi-directional motion was found for the Turkey earthquake. Therefore, the amplification of responses, to a significant extent, can be explained due to the Forward Directivity effect, which should be given prime importance in the seismic design of isolated bridge structures.

Compared to uni-directional Turkey ground motion, the deck acceleration, the force transmitted to the pier, pier torsional moment, and deck displacement increased by about 98%, 95%, 90%, and 102%, respectively, for the bi-directional ground motion.

The increase in maximum responses due to bi-directional ground motion is less for the Kobe earthquake, which is about 10%, 17%, 14%, and 15%, respectively, corresponding to deck acceleration, the force transmitted to the pier, pier torsional moment, and deck displacement. However, as compared to a uni-directional motion, the same responses are enhanced by 16%, 53%, 29%, and 54% for bi-directional Imperial Valley earthquakes.

Figure 15 represents the envelope (corresponding to a 2.5 s time period and 20% damping ratio) of bearing displacement and force transmitted to the pier for Imperial Valley, Kobe, and Turkey earthquake motions. It was noticed that the bi-directional response envelope (red line) surpassed the uni-directional envelope (black line) by a substantial amount. A similar response was observed by Gupta et al. [49–51]. This fact is very important for the design of isolated bridge structures as, in the traditional manner only the principal horizontal direction was considered. However, for actual consideration, the design should have been

governed by the response contribution considering bi-directional loading. In Figure 15, the blue line represents the time history response due to bi-directional loading. As the red line has exceeded both the blue and black lines, the dominance of the response of the bridge structure to the bi-directional motion has been well established.

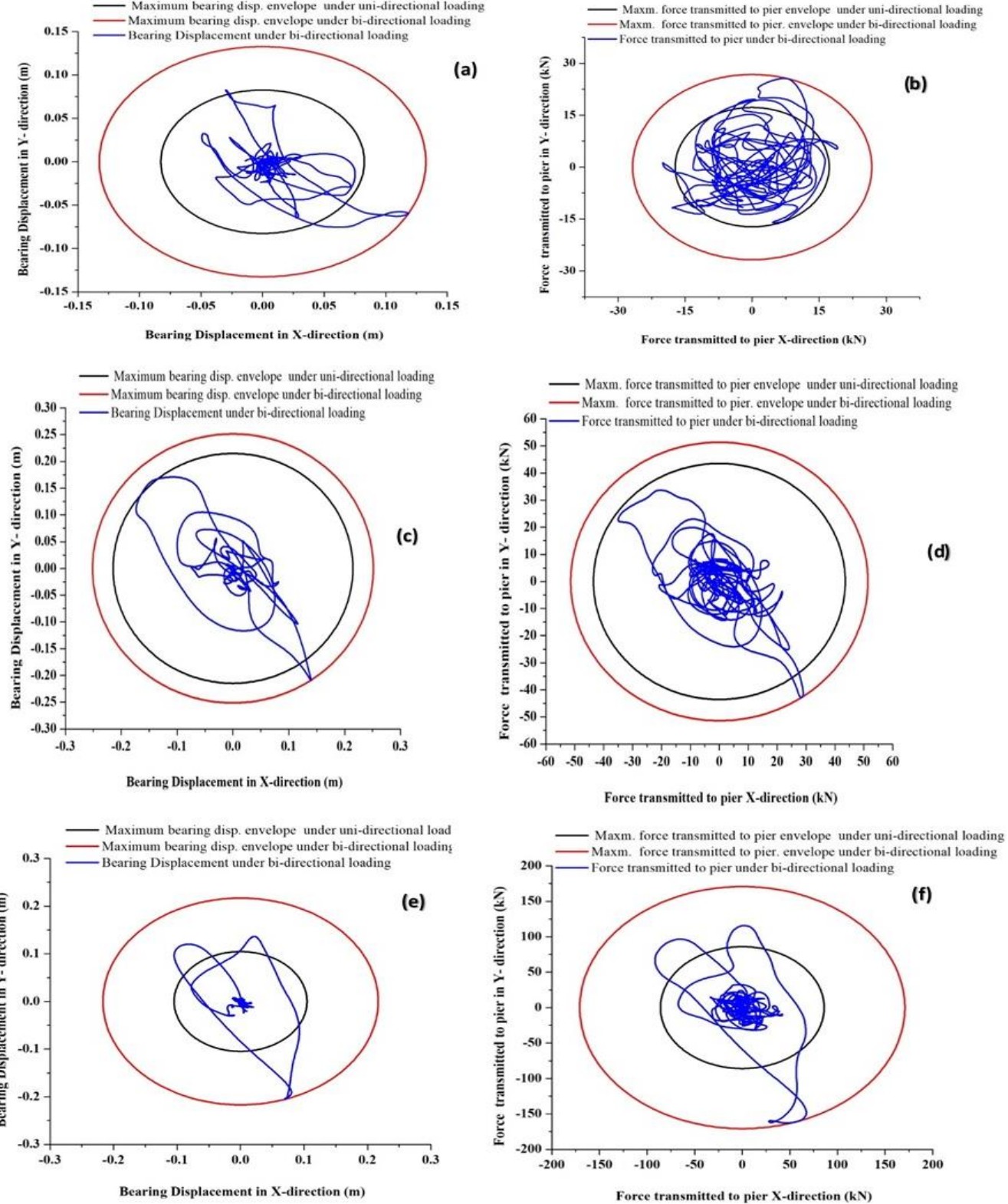

**Figure 15.** Response envelopes of the isolated bridge for Imperial Valley (**a**,**b**), Kobe (**c**,**d**), and Turkey (**e**,**f**) ground motions.

## 8. Conclusions

The efficacy of LRBs as isolation bearings for curved bridges has been examined for various ground motions. The effect of the incidence angle of the earthquake on the uni-directional and bi-directional responses of the bridge was determined. Sensitivity analyses were performed for various time periods and damping ratios of the LRBs. Bi-directional loading effects of ground motions on the response of the bridge were also investigated. The following are the important conclusions of the above study.

LRBs can significantly reduce the curved bridge forces, especially the pier torsional moment, by about 94%, compared to non-isolated conditions. The maximum decrease in forces was found for the Imperial Valley and Kobe earthquakes. In the cases of Imperial Valley and Kobe ground motions, the maximum reduction of force transmitted to the pier was about 84 to 85% for uni-directional cases and 93 to 95% for bi-directional cases, while in the case of Turkey ground motion, the maximum reduction of force transmitted to the pier was about 44 to 71% for uni-directional cases and 56 to 83% for bi-directional cases.

The variations of bridge response are found to be non-linear for design parameters of the LRBs, such as the natural time period and damping ratio, which are supposed to be due to the non-linear force-deformation behavior of the LRBs considered by the coupled plasticity model.

The deck displacement of the LRB isolated bridge was increased, and the maximum effect was observed for Turkey ground motion, especially at higher periods. It is crucial as per the design of isolation bearings.

For most of the response parameters of the bridge, the variations observed were small, except for the pier torsional moment, which was the maximum for an incidence angle of 67.5° to 90°.

The variation of response parameters is more sensitive with respect to the time period than the damping ratio of the bearings.

For various earthquake ground motions, a large variety of bearing displacement with respect to damping ratio was observed for Turkey ground motion, especially at higher time periods, which might be due to the forward directivity effect. However, for bi-directional Turkey earthquake motion, the steepness of variation of bearing displacement has been reduced compared to the same for uni-directional motion.

The pier torsional moment variations are almost insignificant for a higher range of time periods, whereas they are more sensitive at lower time period ranges. In the case of the Imperial Valley earthquake, the maximum variation of pier torsional moment was observed at lower time periods for both uni-directional and bi-directional loadings.

The effect of bi-directional ground motion is found to be crucial as the same has enhanced the response of the bridge by a significant amount. The maximum effect has been found for Turkey ground motion, which has increased the deck acceleration, the force transmitted to the pier, pier torsional moment, and deck displacement by about 98%, 95%, 90%, and 102%, respectively, over the uni-directional case. The results infer the importance of the forward directivity effect in the design of the isolated curved bridge.

**Author Contributions:** Conceptualization, P.K.G. and G.G. Data curation, P.K.G.; Formal analysis, P.K.G. and S.D.; Investigation, P.P.; Methodology, P.K.G. and V.K.; Project administration, P.K.G.; Supervision, V.K.; Validation, P.P.; Visualization, V.K. and S.D.; Writing—original draft, P.K.G., P.P.; Writing—review and editing, Goutam Ghosh S.D. All authors have read and agreed to the published version of the manuscript.

**Funding:** The research work mentioned in this paper was financially supported by the Science and Engineering Research Board (SERB), Department of Science and Technology, New Delhi. Thanks to SERB and the University of Tromso for their support.

**Institutional Review Board Statement:** Not applicable.

**Informed Consent Statement:** Not applicable.

**Conflicts of Interest:** The authors have no relevant financial or non-financial interest to disclose.

## Appendix A

**Table A1.** Uni-directional and bi-directional loadings considered for Imperial Valley ground motion.

| Type | Loading | Direction |
|---|---|---|
| Uni-dir. | 1 | PGA 0.313 g along global long. |
| | 2 | PGA 0.313 g w.r.to 22.5° along global long. |
| | 3 | PGA 0.313 g w.r.to 45° along global long. |
| | 4 | PGA 0.313 g w.r.to 67.5° along global long. |
| | 5 | PGA 0.313 g along global trans. |
| Bi-dir. | 1 | PGA 0.313 g along global long. and PGA 0.215 g along global trans. |
| | 2 | PGA 0.313 g w.r.to 22.5° along global long. and PGA 0.215 g w.r.to 22.5° along global trans. |
| | 3 | PGA 0.313 g w.r.to 45° along global long. and PGA 0.215 g w.r.to 45° along global trans. |
| | 4 | PGA 0.313 g w.r.to 67.5° along global long. and PGA 0.215 g w.r.to 67.5° along global trans. |
| | 5 | PGA 0.313 g w.r.to 90° along global long. and PGA 0.215 g w.r.to 90° along global trans. |

**Table A2.** Uni-directional and bi-directional loadings considered for Kobe ground motion.

| Type | Loading | Direction |
|---|---|---|
| Uni-dir. | 1 | PGA 0.83 g along global long. |
| | 2 | PGA 0.83 g w.r.to 22.5° along global long. |
| | 3 | PGA 0.83 g w.r.to 45° along global long. |
| | 4 | PGA 0.83 g w.r.to 67.5° along global long. |
| | 5 | PGA 0.313 g along global trans. |
| Bi-dir. | 1 | PGA 0.83 g along global long. and PGA 0.63 g along global trans. |
| | 2 | PGA 0.83 g w.r.to 22.5° along global long. and PGA 0.63 g w.r.to 22.5° along global trans. |
| | 3 | PGA 0.83 g w.r.to 45° along global long. and PGA 0.63 g w.r.to 45° along global trans. |
| | 4 | PGA 0.83 g w.r.to 67.5° along global long. and PGA 0.63 g w.r.to 67.5° along global trans. |
| | 5 | PGA 0.83 g w.r.to 90° along global long. and PGA 0.63 g w.r.to 90° along global trans. |

**Table A3.** Uni-directional and bi-directional loadings considered for Turkey ground motion.

| Type | Loading | Direction |
|---|---|---|
| Uni-dir. | 1 | PGA 0.5 g along global long. |
| | 2 | PGA 0.5 g w.r.to 22.5° along global long. |
| | 3 | PGA 0.5 g w.r.to 45° along global long. |
| | 4 | PGA 0.5 g with 67.5° along global long. |
| | 5 | PGA 0.5 g along global trans. |
| Bi-dir. | 1 | PGA 0.5 g along global long. and PGA 0.4 g along global trans. directions |
| | 2 | PGA 0.5 g w.r.to 22.5° along global long. and PGA 0.4 g w.r.to 22.5° along global trans. |
| | 3 | PGA 0.5 g w.r.to 45° along global long. and PGA 0.4 g w.r.to 45° along global trans. |
| | 4 | PGA 0.5 g w.r.to 67.5° along global long. and PGA 0.4 g w.r.to 67.5° along global trans. |
| | 5 | PGA 0.5 g w.r.to 90° along global long. and PGA 0.4 g w.r.to 90° along global trans. |

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
