# Peer review of "Effectiveness of LRB in Curved Bridge Isolation: A Numerical Study"

_applsci, doi:10.3390/app122111289_

Round 1

Reviewer 1 Report

This paper takes the seismic response of the curved continuous isolation bridge using LRB as the object, the seismic response of the bridge under three different types of ground motion input as well as the effectiveness of LRB in curved bridge isolation being investigated via numerical analysis method. The results provide some references for the seismic isolation design of curved bridges. However, the paper has the following problems:

1) LRB is already a very mature technology for isolation device. The innovation of the paper is not obvious enough.

2) In the numerical analysis of seismic response of curved isolation bridges considering horizontal bi-directional seismic deformation, the hysteretic model of LRB become to more complex than that in unidirectional motion. In this study, it is very important to adopt a reliable LRB bi-directional deformation hysteretic model. However, only formulas (4) and (5) are given in the paper, and the parameters of the hysteresis model and the interaction between the two directions of deformation are not described in detail.

3) The seismic responses in this paper are obtained based on merely an example, so that the results seem not universal. If the structural parameters change, the calculation results will also change. Therefore, as long as the representative results are retained, most of the calculation results in the paper can be omitted.

4) In Introduction, the elaboration of [15-22] is repeated.

Author Response

The authors are grateful to the reviewers for thorough review, very constructive suggestions and

corrections in the language/text. All the suggestions of reviewers have been incorporated in the

revised manuscript. The detailed response to specific comments/suggestions has been provided

below:

Reviewer 2 Report

Comments and suggestions in the attached pdf file.

Author Response

Comments and suggestions has been incorporated in word file with track changes options.

The authors are grateful to the reviewers for thorough review, very constructive suggestions and

corrections in the language/text. All the suggestions of reviewers have been incorporated in the

revised manuscript. The detailed response to specific comments/suggestions has been provided in the new file.

Reviewer 3 Report

This paper attempts to provide a numerical investigation of using LRB for curved bridge subjected to seismic load. The paper has serious problem in its English level. I have read abstract many times but I could not understand it well. when I started to read the introduction, I saw that the first paragraph of the introduction has more problems in its English level. I cannot understand and provide proper comments for a paper that I cannot understand it well.

I give up from reading the rest of it and I reject it. I strongly suggest the authors to let a native revises their paper and submit a revised version of their paper again. Additionally please consider two small comments as below:

1-    All abbreviation should be defined in the paper, exactly before using the abbreviation. For example, in the first line of the abstract, LRB is not defined in advance.

2-    Please use the same format in your write. In abstract you wrote “Bi-directional” and “Bidirectional”. Choose one of them. Check this for other words in entire the paper.

Author Response

This paper attempts to provide a numerical investigation of using LRB for curved bridge subjected to seismic load. The paper has serious problem in its English level. I have read abstract many times but I could not understand it well. when I started to read the introduction, I saw that the first paragraph of the introduction has more problems in its English level. I cannot understand and provide proper comments for a paper that I cannot understand it well.

I give up from reading the rest of it and I reject it. I strongly suggest the authors to let a native revises their paper and submit a revised version of their paper again. Additionally please consider two small comments as below:

Response: The paper has been revised as per the suggestions of the reviewer.

1- All abbreviation should be defined in the paper, exactly before using the abbreviation. For example, in the first line of the abstract, LRB is not defined in advance.

Response: The first line of the abstract has been modified as suggested by reviewers.

2- Please use the same format in your write. In abstract you wrote “Bi-directional” and “Bidirectional”.

Choose one of them. Check this for other words in entire the paper.

Response: The author has considered “Bi-directional” and used the same format throughout the papery.

Round 2

Reviewer 2 Report

The authors made the required corrections thus the paper can be accepted.

Reviewer 3 Report

the comments are attached
